# Utility of ctDNA Liquid Biopsies from Cancer Patients: An Institutional Study of 285 ctDNA Samples

**DOI:** 10.3390/cancers14235859

**Published:** 2022-11-28

**Authors:** Josep Gumà, Karla Peña, Francesc Riu, Carmen Guilarte, Anna Hernandez, Clara Lucía, Francisca Martínez-Madueño, Maria José Miranda, Inés Cabezas, Marc Grifoll, Sergio Peralta, Sara Serrano, Félix Muñoz, Lola Delamo, Barbara Roig, Joan Borràs, Joan Badia, Marta Rodriguez-Balada, David Parada

**Affiliations:** 1Institut d’Oncologia de la Catalunya Sud, Hospital Universitari Sant Joan de Reus, IISPV, URV, 43204 Reus, Spain; 2Institut d’Investigació Sanitària Pere Virgili, 43204 Reus, Spain; 3Facultat de Medicina i Ciències de la Salut, Universitat Rovira i Virgili, 43204 Reus, Spain; 4Molecular Pathology Unit, Department of Pathology, Hospital Universitari de Sant Joan, 43204 Reus, Spain

**Keywords:** liquid biopsy, ctDNA, cancer, lung, colorectal, melanoma, precision medicine

## Abstract

**Simple Summary:**

Liquid biopsy is a minimally invasive complementary tool useful in cancer patients for the early identification of treatment selection and resistance to cancer treatment, as well as in routine cancer follow-up. Due to its potential, it is necessary to promote and assess its implementation in the practice of precision medicine.

**Abstract:**

Liquid biopsy has improved significantly over the last decade and is attracting attention as a tool that can complement tissue biopsy to evaluate the genetic landscape of solid tumors. In the present study, we evaluated the usefulness of liquid biopsy in daily oncology practice in different clinical contexts. We studied ctDNA and tissue biopsy to investigate *EGFR*, *KRAS*, *NRAS*, and *BRAF* mutations from 199 cancer patients between January 2016 and March 2021. The study included 114 male and 85 female patients with a median age of 68 years. A total of 122 cases were lung carcinoma, 53 were colorectal carcinoma, and 24 were melanoma. Liquid biopsy was positive for a potentially druggable driver mutation in 14 lung and colorectal carcinoma where tissue biopsy was not performed, and in two (3%) lung carcinoma patients whose tissue biopsy was negative. Liquid biopsy identified nine (45%) de novo *EGFR-T790M* mutations during TKI-treatment follow-up in lung carcinoma. *BRAF-V600* mutation resurgence was detected in three (12.5%) melanoma patients during follow-up. Our results confirm the value of liquid biopsy in routine clinical oncologic practice for targeted therapy, diagnosis of resistance to treatment, and cancer follow-up.

## 1. Introduction

In the era of precision oncology, new technologies need to be implemented to identify the molecular markers that, thanks to the somatic characterization of the alterations involved in tumor progression, can be used as prognostic and predictive factors in daily practice. Tissue biopsy is currently regarded as the “gold standard” in diagnosis for cancer treatment [1]. However, the inherent limitations of tissue biopsy do not allow it to be routinely used in cancer monitoring, since it is more useful for providing specific information than for demonstrating the dynamic evolutionary characteristics of cancer [2,3]. In addition, the difficulty in obtaining tumor tissue samples from areas that are difficult to access or from patients whose clinical condition is deteriorating is a daily challenge in oncology practice and may have implications for decision-making.

Liquid biopsy is a technology that has improved significantly over the last decade and is attracting attention as a tool that can complement tissue biopsy because it is minimally invasive and enables the genetic landscape of solid tumors to be evaluated [4,5,6,7,8]. Currently, liquid biopsy is used in oncology primarily to study mutations in circulating tumor DNA (ctDNA). Thus, in lung cancer, ctDNA enables the detection of mutations that are potential therapeutic targets, such as epidermal growth factor receptor (EGFR) mutations. In addition, ctDNA allows one to demonstrate the emergence of mutations involved in mechanisms of resistance to tyrosine kinase inhibitor (TKI) therapy, such as the epidermal growth factor receptor (*EGFR*) mutation. Additionally, liquid biopsy allows the detection of deletions or mutations in other pathways, such as on anaplastic lymphoma kinase (*ALK)*, c-Ros oncogene 1 (*ROS1*), MET proto-oncogene (*MET*), and B-Raf proto-oncogene (*BRAF*) by NGS techniques, whose manifestation allows the application of specific treatments.

This study was performed with the objective of assessing the usefulness of ctDNA liquid biopsy (ctDNALB) in patients with different oncological diseases, taking into consideration three key clinical aspects: the ability of liquid biopsy to provide information on resistance to treatment, to predict response to treatment, and its usefulness in the follow-up of cancer patients.

## 2. Materials and Methods

### 2.1. Patient Selection

This is a retrospective and descriptive cohort study conducted on patients with lung or colorectal carcinomas, or melanoma under follow-up care, at the Medical Oncology Service of the South Catalonia Oncology Institute (Hospital Universitari de Sant Joan, Reus, Spain) between 1 January 2016 and 31 March 2021. We studied blood samples from 200 patients submitted to the molecular pathology unit of our pathology department. The patients’ clinical data were extracted from medical records.

### 2.2. ctDNA Liquid Biopsy Processing for Mutation Determination by the IdyllaTM System

From each patient, 10 mL of venous blood was collected in a cell-free DNA BCT^®^ (Leipzig, Germany) tube and gently homogenized at least 10 times. The blood sample was immediately sent to the pathology department, where it was centrifuged for 10 min at 1600× *g*. The centrifugation process was progressively stopped. After centrifugation, the plasma was carefully extracted while always avoiding contact with the buffy coat. The plasma was aliquoted in a volume of 1.2 mL and centrifuged for 10 min at 6000× *g* in a microcentrifuge. The supernatant plasma was carefully removed for ctDNA analysis, avoiding the debris pellet. Plasma aliquots (1.1 mL) were prepared. Finally, to investigate *EGFR*, *KRAS*, *NRAS*, and *BRAF* mutations in ctDNA, 1 mL of plasma was placed into IdyllaTM *EGFR*, *KRAS*, *NRAS*, and *BRAF* mutation cartridges for a fully automated assay. The remaining plasma samples from each patient were stored at −80 °C as backup. To assess the clinical application of ctDNA liquid biopsies, the following aspects were analyzed: (1) its benefit as a marker of response to treatment; (2) its ability to detect resistance to treatment; and (3) its usefulness in follow-up care. The *EGFR*, *KRAS*, *NRAS*, and *BRAF* mutations detected in the IdyllaTM *EGFR*, *KRAS*, *NRAS*, and *BRAF* mutation cartridges are listed in Appendix A.

### 2.3. Tissue Biopsy Processing by the IdyllaTM System for Mutation Determination

In 144 patients, tumor tissue biopsy samples were obtained and subjected to mutation tests for *EGFR*, *KRAS*, *NRAS*, and *BRAF* in the IdyllaTM system (Mechelen, Belgium), according to the manufacturer’s recommendations. In short, 20-micron-thick sections were obtained from the paraffin-embedded material and placed in cartridges to study *EGFR*, *KRAS*, *NRAS*, and *BRAF* tissue mutations using fully automated testing. Following the manufacturer’s recommendations for the study of the mutation, in the case of EGFR, a minimum proportion of tumor cells in the biopsy of 10% was required. For the rest of the mutations, a minimum proportion of tumor cells greater than 50% was required. In some cases, in which the proportion of tumor cells did not meet the quantitative parameters, microsections of the neoplasm were made in order to increase the sufficiency of material to carry out the study. The *EGFR*, *KRAS*, *NRAS*, and *BRAF* mutations detected in the IdyllaTM *EGFR*, *KRAS*, *NRAS*, and *BRAF* mutation cartridges are listed in Appendix A.

## 3. Results

### 3.1. Clinical Findings

Our analysis included 114 (57%) male and 85 (43%) female patients with a median age of 68 years (range, 60–75 years). A total of 122 cases (61%) had lung carcinoma, 53 (27%) had colorectal carcinoma, and 25 (12%) had melanoma. Clinical stage IV was predominant in 166 (83%) patients, followed by stage III in 30 patients (15%) (Table 1). The most common histological type was adenocarcinoma in 150 (75%) patients, followed by melanoma in 27 (14%) patients (Table 1).

### 3.2. ctDNA Liquid Biopsy for Treatment Selection

From 157 patients with lung or colorectal cancer, 29 patients (18%) showed positive ctDNA liquid biopsies, with genetic mutations in *EGFR* (*n* = 9), *BRAF* (*n* = 3), neuroblastoma rat sarcoma virus (*NRAS)* (*n* = 1), and Kirsten rat sarcoma virus (*KRAS*) (*n* = 16). In 16 patients out of 29, in whom tissue biopsy was negative or not available (2 and 14 patients, respectively), ctDNA liquid biopsy demonstrated actionable genetic mutations (Table 2). Segregated by tumor location, there were five positive ctDNA liquid biopsies out of nine that had no tissue biopsy available for lung cancer and nine positive ctDNA liquid biopsies out of 20 that had no tissue biopsy available for colorectal cancer. Additionally, there were two positive ctDNA liquid biopsies with lung cancer that had a negative tissue biopsy (Table 2). Focusing on genes—*BRAF*-, *KRAS*- and *NRAS*-positive ctDNA liquid biopsies—matched the results of tissue biopsy when it was available, while there were two cases in which *EGFR* was positive in liquid biopsy, but negative in tissue biopsy (Table 3).

### 3.3. ctDNA Liquid Biopsy and Resistance to Oncological Treatment

At this point, the study focused on the detection of genetic mutations involved in treatment resistance, specifically the *EGFR*-T790M mutation in lung adenocarcinoma. The patients enrolled in this part of the study (*n* = 20) had an initial *EGFR* mutation and were eligible for anti-*EGFR* targeted therapy, and ctDNA liquid biopsies (*n* = 37) were performed during treatment follow-up to monitor treatment resistance. Of the 20 patients enrolled with an initial *EGFR* mutation, nine (45%) presented a positive *EGFR*-T790M liquid biopsy during treatment follow-up. Considering the initial *EGFR* mutation, four of the 11 (36%) *EGFR*-DEL19 were positive for *EGFR*-T790M, and five of the eight (62%) *EGFR*-L858R were positive for *EGFR*-T790M (Table 4).

### 3.4. ctDNA Liquid Biopsy in Follow-Up Care

Melanoma patients with an initial mutation in the *BRAF* gene (*n* = 24) were recruited for follow-up with ctDNA liquid biopsies (*n* = 91). Almost all of them (*n* = 22, 92%) had a negative ctDNA liquid biopsy during follow-up. Additionally, the initial mutation was redetected in three patients, meaning a positive ctDNA liquid biopsy followed a negative previous ctDNA liquid biopsy.

Lung adenocarcinoma patients with an initial mutation in the *EGFR* gene (*n* = 20) were enrolled for follow-up with ctDNA liquid biopsies (*n* = 37). Half of them (*n* = 11, 55%) had a negative ctDNA liquid biopsy during follow-up. The initial mutation was redetected in only one patient.

## 4. Discussion

Currently, tumor tissue analysis is the gold standard for diagnosing cancer. However, its use in clinical practice is limited by the lack of tissue material amenable to biopsy, the clonal heterogeneity of the tumor, and the risk of surgical complications resulting from multiple biopsy procedures. This increases the need for new methods, such as liquid biopsy, which, in addition to being minimally invasive, can be used to study different aspects of a tumor [4,5,6,7,8]. It is interesting to carry out these types of retrospective institutional studies to prove the efficiency and effectiveness of liquid biopsy to analyze the presence of mutations in real-world data in cancer patients that could change the medical decisions in daily practice.

In lung cancer, especially in the adenocarcinoma subtype, the detection of a mutation in the *EGFR* gene is extremely important for choosing the first line of treatment. *EGFR*-targeted therapies, such as erlotinib and gefitinib, have shown great efficacy in the treatment of these patients. Currently, *EGFR* inhibitors are the first-line chemotherapy treatment in patients with EGFR-mutant tumors [9,10]. To detect these *EGFR* gene mutations and select the optimal treatment, we performed a total of 104 liquid biopsies in patients diagnosed with lung cancer in our daily clinical practice. In these 104 ctDNA samples, we detected nine *EGFR* mutations, indicating that approximately 9% of patients had the mutation. These data resemble those provided in the literature on the prevalence of this genetic alteration in populations similar to that of our study [11]. A key point in this study is that five (15.2%) of these *EGFR* mutations have been detected in patients where it was not possible to perform the mutational study in tissue, either because of the clinical situation or because of tumor locations not accessible to biopsy. The importance of this lies in the fact that liquid biopsy allows us to carry out an essential study for the treatment of patients in whom it would not have been possible before. We would also note that two (3%) of the *EGFR* mutations were present in the liquid biopsy, but not in the tissue sample. This highlights the great usefulness of using liquid biopsy in patients where the mutation was not detected in the tissue sample, but once they have progressed to chemotherapy treatment, a new determination is made in peripheral blood, and the mutation is obtained. This can be explained for two reasons: one of them is the limitations of tissue biopsy to detect mutations given the tumor heterogeneity, another reason is that tumor evolution leads to the addition of new mutations that are not present at the beginning and that noninvasive techniques, such as liquid biopsy, allow us to discard them quickly and efficiently. So, in oncological clinical practice, it is essential to detect actionable mutations in those patients who are eligible to receive targeted drugs. In our study, liquid biopsy would have made it possible to detect up to 3% of patients with an *EGFR* mutation that had not been identified in the tissue biopsy because of tumor heterogeneity and tumor evolution. In any case, the sensitivity of liquid biopsy to detect an *EGFR* mutation in our study is similar to that of tissue biopsy, as has been demonstrated in other studies [12].

In metastatic CRC (mCRC), mutations in the *RAS* oncogene family (especially *KRAS* and *NRAS* genes) or *BRAF* genes are related to poor disease prognosis [13] and are negative predictive factors for the response to anti-*EGFR* therapy (cetuxiamb or panitumumab) [14,15,16,17]. Given the importance of detecting these mutations to choose the best treatment, we performed a total of 53 liquid biopsies. We found a total of 20 mutations: 17 in *KRAS/NRAS* and three in *BRAF*. Once again, it is noteworthy that nine (37.5%) of these mutations in *KRAS/NRAS* were detected in patients for whom we did not have tissue studies due, on many occasions, to the scarce endoscopic tissue sample obtained or the presence of significant necrosis in the tissue, highlighting once again the great usefulness of this technique in daily clinical practice where we do not always have tissue mutation studies available. It should be noted that, in all patients in whom both tissue and liquid biopsies were available, the results were identical in both scenarios.

Therefore, the analysis of *EGFR* and *RAS/BRAF* mutations in plasma using the IdyllaTM platform is feasible and reliable in clinical practice and allows the selection of the best treatment for our patients, especially those with metastatic lung adenocarcinomas and colon cancer. In addition, it allows the study of mutations in those patients for whom we do not have a tissue sample, making it an indispensable technique in our clinical practice today.

Another important aspect of liquid biopsy is that it allows the detection of resistance mutations associated with primary and acquired resistance during TKI treatment without the need for new biopsies that rely on invasive techniques. For patients who progress on a first- or second-generation *EGFR* TKI, the guidelines recommend repeated tumor or liquid biopsy to identify mechanisms of acquired resistance. The most frequent mechanism of acquired resistance to first- and second-generation *EGFR*-TKIs in *EGFR*-mutated lung cancer is the presence of an *EGFR T790M* mutation in exon 20 and confers sensitivity to third-generation inhibitors [18,19]. Of the 20 patients enrolled with an initial EGFR mutation, nine (45%) presented a positive *EGFR T790M* liquid biopsy during TKI-treatment follow-up, which led to their treatment being changed. Therefore, the *T790M EGFR* resistance mutation was found in 45% of our cohort, while other studies have shown lower rates of *EGFR T790M* mutation using the same detection system [20,21]. This difference can be explained by the total number of patients in the previous study and the follow-up time. In essence, the high prevalence of this resistance mutation in our cohort demonstrates the great usefulness of liquid biopsy in this scenario and the need to apply this technique to these patients in our daily clinical practice.

Early detection of relapses in the follow-up of patients undergoing treatment with targeted drugs is a situation that can benefit from the use of liquid biopsy, especially in patients with advanced melanoma. Treatment for melanoma has dramatically changed over the past decade with the introduction of immune checkpoint inhibitors and BRAF, plus MEK-targeted therapies into clinical practice [22,23]. Various studies have shown that liquid biopsy can be used to detect mutated BRAF, which can be identified in 32% of stage I/II patients and 39% of stage III/IV patients [24,25,26,27,28,29]. Patients with *BRAFV600*-mutant tumors can be treated with highly effective *BRAF* plus *MEK*-targeted therapies (dabrafenib/trametinib, vemurafenib/cobimetinib, and encorafenib/binimetinib) with significant overall survival benefits [23]. Assessment of *BRAFV600* mutant circulating cell-free tumor DNA could be a tool for therapeutic monitoring in *BRAFV600* mutant metastatic melanoma patients treated with *BRAF/MEK* inhibitors, since a relationship has been shown between this ctDNA, durable responses, and survival outcomes [30]. We studied a total of 24 advanced melanoma patients with an initial tissue mutation in the BRAF gene, and we performed a total of 92 ctDNA liquid biopsies during follow-up. Five (21%) of these patients had an initial BRAF mutation-positive liquid biopsy. Under treatment, the initial mutation was redetected in three patients (16%), meaning a positive ctDNA liquid biopsy followed a prior negative ctDNA liquid biopsy. These findings will allow us to anticipate a possible relapse to treatment. Our results support the usefulness of ctDNA in the follow-up care of patients with melanoma since it enables relapse to be identified early. It should be noted that ctDNA is increasingly being utilized for tumor response monitoring and identification of minimal residual disease, but in our case we are using a qualitative determination of the BRAF mutation as a first step to detect an early recurrence of the disease, not a quantitative determination that allows us to monitor minimal residual disease or quantify the presence of residual disease [31].

Despite not being performed routinely in clinical practice, 20 patients diagnosed with lung adenocarcinoma with an initial tissue mutation in the *EGFR* gene were also followed up to fine-tune this technique. A total of 37 liquid biopsies were performed. Notably, in half of these patients, the mutation was not detected in the liquid biopsy during follow-up and reappeared in one of them. This would allow us, as in the case of melanoma, to anticipate a possible relapse by means of this technique, thus complementing the usual follow-up tests required for these patients.

Finally, it should be noted that the study included three patients in whom *EGFR* detection was performed for treatment selection despite being diagnosed with stage II lung adenocarcinoma. In these patients, where the result of the liquid biopsy was negative, no tissue study was performed because they were not candidates for invasive diagnostic studies or for chemotherapy, radiotherapy, or surgery due to age and/or comorbidities. Therefore, their only treatment options were targeted therapy or immunotherapy. In this way, liquid biopsy, as a noninvasive, safe, and easy procedure, has the potential to improve the currently used strategies for lung cancer diagnosis and treatment, either alone or as complementary data for imaging findings in this type of patient [32]. Nevertheless, further studies in this direction are needed to make the use of liquid biopsy in this setting a reality.

Our study has some limitations: (a) the sample includes different types of solid cancers. This raises the possibility that the differences in cell turnover between the different types of tumors, their histopathological characteristics and their biological behavior confer differences in the sensitivity of liquid biopsy; (b) The concentration of ctDNA can vary according to the tumor biorhythm, so the samples for the study of ctDNA can present plasmatic variations. In our patients, the extraction of the blood sample for the liquid biopsy was performed between 8 AM and 3 PM, but we do not have more precise data; and (c) because samples were taken in advanced stages, conclusive data on early mutational changes could not be obtained.

## 5. Conclusions

In conclusion, this study demonstrated that circulating tumor DNA can be safely implemented in clinical practice in cancer patients in three crucial aspects of oncological disease: (1) as a predictive biomarker of response to targeted therapy and, consequently, for the selection of treatment; (2) as a method for diagnosing resistance to treatment; and (3) as a method for cancer follow-up. This procedure is noninvasive and allows real-time study of the heterogeneity of the tumor and its implications for patient survival. However, prospective studies are required to assess the sensitivity, specificity, cost, and tumor circadian rhythm.

## Figures and Tables

**Table 1 cancers-14-05859-t001:** Clinical characteristics of cancer patients under investigation for ctDNA liquid biopsy test (*n* = 199).

	Total	Treatment Selection	Treatment Resistance	Disease Follow-Up
Patients	199	157	20	24
Liquid biopsies	285	157	37	91
Age, median (Q1, Q3)	68 (60, 75)	69 (63, 76)	62 (59, 71)	59 (49, 68)
Sex, *n* (%)				
Male	114 (57%)	96 (61%)	7 (35%)	12 (50%)
Female	85 (43%)	61 (39%)	13 (65%)	12 (50%)
Stage, *n* (%)				
II	3 (2%)	3 (2%)	0	0
III	30 (15%)	12 (8%)	0	18 (75%)
IV	165 (83%)	141 (90%)	20 (100%)	6 (25%)
Not available	1 (<1%)	1 (<1%)	0	0
Tumor location, *n* (%)				
Colorectal	53 (27%)	53 (34%)	0	0
Lung	122 (61%)	104 (66%)	20 (100%)	0
Melanoma	24 (12%)	0	0	24 (100%)
Histology, *n* (%)				
SCLC	1 (0.5%)	1 (1%)	0	0
NSCLC	5 (2.5%)	5 (3%)	0	0
Adenocarcinoma	150 (75%)	132 (84%)	20 (100%)	0
Large cell neuroendocrine carcinoma	2 (1%)	2 (1%)	0	0
Squamous	10 (5%)	10 (6%)	0	0
Melanoma	26 (13%)	2 (1%)	0	24 (100%)
Not available	5 (2.5%)	5 (3%)	0	0

Abbreviations: SCLC: Small cell lung carcinoma; NSCLC: Non-small cell lung carcinoma.

**Table 2 cancers-14-05859-t002:** Lung and colorectal cancer patients under investigation for ctDNA liquid biopsy test and treatment selection.

	Liquid Biopsy Positive	Liquid Biopsy Negative
Overall (*n* = 157)	28	129
Tissue biopsy positive	12	6
Tissue biopsy negative	2	80
No tissue biopsy	14	43
Lung (*n* = 104)	9	95
Tissue biopsy positive	2	2
Tissue biopsy negative	2	65
No tissue biopsy	5	28
Colorectal cancer (*n* = 53)	19	34
Tissue biopsy positive	10	4
Tissue biopsy negative	0	15
No tissue biopsy	9	15

**Table 3 cancers-14-05859-t003:** Lung and colorectal cancer patients under investigation for genomic mutations identified with ctDNA liquid biopsy and treatment selection.

Tumor	Total Patients (*n*, %)	Tissue Biopsy Positive (*n*, %)	Tissue Biopsy Negative (*n*, %)	No Tissue Biopsy (*n*, %)
Lung	104	4	67	33
LB− total	95 (91.3%)	2 (50%)	65 (97%)	28 (84.8%)
LB+ total	9 (8.7%)	2 (50%)	2 (3%)	5 (15.2%)
*EGFR-DEL19*	3 (2.9%)	1 (25%)	0	2 (6.1%)
*EGFR-L858R*	6 (5.8%)	1 (25%)	2 (3%)	3 (9.1%)
Colorectal	53	14	15	24
LB− total	34 (64.2%)	4 (28.6%)	15 (100%)	15 (62.5%)
LB+ total	19 (35.8%)	10 (71.4%)	0	9 (37.5%)
*KRAS G12C*	2 (3.8%)	0	0	2 (8.3%)
*KRAS G12D*	5 (9.4%)	3 (27.3%)	0	2 (8.3%)
*KRAS G12S*	1 (1.9%)	0	0	1 (4.2%)
*KRAS G12V*	3 (7.5%)	1 (9.1%)	0	3 (12.5%)
*KRAS G13D*	3 (5.7%)	3 (27.3%)	0	0
*KRAS NOS*	1 (1.9%)	0	0	1 (4.2%)
*BRAF V600E/D*	3 (5.7%)	3 (27.3%)	0	0

**Table 4 cancers-14-05859-t004:** Lung adenocarcinoma patients under investigation for genomic mutations identified with ctDNA liquid biopsy and detection of EGFR-T790M during follow-up care.

	Initial Alteration *n*	Follow-Up *EGFR*-T790M *n* (%)
*EGFR* mutations	20	9 (45)
*EGFR*-DEL19 (exon 19)	11	4 (36)
*EGFR*-L858R (exon 21)	8	5 (62)
*EGFR*-G719X (exon 18)	1	0 (0)

## Data Availability

All study data are included.

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
