# Peer review of "Utility of ctDNA Liquid Biopsies from Cancer Patients: An Institutional Study of 285 ctDNA Samples"

_cancers, 2022, doi:10.3390/cancers14235859_

Round 1
Reviewer 1 Report (Previous Reviewer 2)
I think the revisions do point out the numeric rates of potential benefits to performing liquid bx even occasionally when tumor negative for a given mutation so as not to miss druggable mutations.
I think the improved details provided help to overcome the weakness in terms of novelty of the findings.
Reviewer 2 Report (Previous Reviewer 1)
After changes were made, I find the manuscript acceptable for publication.
This manuscript is a resubmission of an earlier submission. The following is a list of the peer review reports and author responses from that submission.
Round 1
Reviewer 1 Report
In this work, the usefulness of ctDNA liquid biopsies in the daily oncology practice of patients with lung carcinoma, colorectal carcinoma, or melanoma was assessed.
The manuscript is well-structured and contains some novel data. However, there are a few issues that need to be clarified.
In section 2.2. it is written that ‘’IdyllaTM EGFR, KRAS, NRAS and BRAF mutation cartridges are listed in Supplementary Table S1‘’, while in section 2.3. ‘’mutations detected in the IdyllaTM EGFR, KRAS, NRAS and BRAF mutation cartridges are listed in Supplementary Table S2‘’. However, I only found Table S1 in the supplementary file attached to the manuscript.
The following part in section 2.3. is not fully understandable for researchers not familiar with IdyllaTM system:
‘’According to the tumor and the mutation studied, a minimum of 10% viable tumor was required in the case of EGFR, while for the rest of the mutations, the minimum volume required was more than 50%. In some cases, the sample was enriched to obtain enough material to carry out the study.‘’.
What is meant by ‘’10% viable tumor’’ or ‘’volume required more than 50%’’? The volume of 50% also refers to viable tumor? How was the sample enriched if there was no enough material?
In the Results section, it would be more informative if there was a statement about Tables 2 and 3 that liquid biopsy positive results for lung carcinoma all refer to EGFR, and BRAF, KRAS/NRAS all refer to colorectal carcinoma (and melanoma was not included in these tables).
In Table 1 there are 25 melanoma patients, and in the Discussion section, lines 230 and 231 say ‘’We studied a total of 24 advanced melanoma patients… and we performed a total of 92 ctDNA liquid biopsies during follow-up‘’, while in Results section it is written ‘’Melanoma patients with an initial mutation in the BRAF gene (n=24) were recruited for follow-up with ctDNA liquid biopsies (n=91)‘’. I suggest these numbers should be revised and harmonized.
Since this manuscript contains data demonstrating that ctDNA can be implemented into clinical practice in cancer patients, I suggest it can be accepted for publication after minor revision.
Reviewer 2 Report
The authors present data from a single institution re: findings of ctDNA assays with /or in lieu of tissue biopsies from the same patients. They demonstrate the utility in identifying driver mutations in lung and colon cancer pts, and melanoma pts. The data and findings are not particularly novel and other authors have published detailed reviews and similar findings. This paper presentation might be more helpful if the percentages for finding driver mutations in liquid bx (+) vs tissue biopsy (-)were specifically calculated and emphasized as this would be an important finding, as practitioners do not want to miss a potentially druggable mutation. The use of ctDNA is increasingly being utilized for tumor response monitoring, identification of minimal residual disease, and resistance as mentioned. Thus, these findings are not particularly novel either.